# Local Recurrence after Endoscopic Submucosal Dissection of Early Gastric Cancer

**DOI:** 10.3390/jcm12052018

**Published:** 2023-03-03

**Authors:** Dae-Gon Ryu, Su-Jin Kim, Cheol-Woong Choi, Su-Bum Park, Hyeong-Seok Nam, Si-Hak Lee, Sun-Hwi Hwang

**Affiliations:** 1Department of Internal Medicine, Pusan National University School of Medicine and Research Institute for Convergence of Biomedical Science and Technology, Pusan National University Yangsan Hospital, Yangsan 50612, Republic of Korea; 2Department of Surgery, Pusan National University School of Medicine and Research Institute for Convergence of Biomedical Science and Technology, Pusan National University Yangsan Hospital, Yangsan 50612, Republic of Korea

**Keywords:** early gastric cancer, endoscopic submucosal dissection, recurrence, scar

## Abstract

Endoscopic submucosal dissection (ESD) is considered the treatment of choice for early gastric cancer (EGC) with a negligible risk of lymph node metastasis. Locally recurrent lesions on artificial ulcer scars are difficult to manage. Predicting the risk of local recurrence after ESD is important to manage and prevent the event. We aimed to elucidate the risk factors associated with local recurrence after ESD of EGC. Between November 2008 and February 2016, consecutive patients (n = 641; mean age, 69.3 ± 9.5 years; men, 77.2%) with EGC who underwent ESD at a single tertiary referral hospital were retrospectively analyzed to evaluate the incidence and factors associated with local recurrence. Local recurrence was defined as the development of neoplastic lesions at or adjacent to the site of the post-ESD scar. En bloc and complete resection rates were 97.8% and 93.6%, respectively. The local recurrence rate after ESD was 3.1%. The mean follow-up period after ESD was 50.7 ± 32.5 months. One case of gastric cancer-related death (0.15%) was noted, wherein the patient had refused additive surgical resection after ESD for EGC with lymphatic and deep submucosal invasion. Lesion size ≥15 mm, incomplete histologic resection, undifferentiated adenocarcinoma, scar, and the absence of erythema of the surface were associated with a higher risk of local recurrence. Predicting local recurrence during regular endoscopic surveillance after ESD is important, especially in patients with a larger lesion size (≥15 mm), incomplete histologic resection, surface changes of scars, and no erythema of the surface.

## 1. Introduction

Early gastric cancer (EGC) is an adenocarcinoma confined within the mucosa or submucosa, regardless of lymph node metastasis. In recent years, many EGC lesions have been removed using endoscopic maneuvers. Among endoscopic resection techniques, endoscopic submucosal dissection (ESD) has been accepted as the treatment of choice for EGC without lymph node metastasis. Shorter operation time, a shorter period of hospital stay, lower costs, and a better quality of life are advantages over surgical resection [1]. With the development of various endoscopic equipment such as electrosurgical knives, electrosurgical units, and suture devices, ESD has achieved higher en bloc and complete pathologic resection rates than conventional endoscopic mucosal resection (EMR) with an electrosurgical snare [2].

In South Korea, the National Cancer Screening Program for gastric cancer includes performing biennial endoscopic examinations or radiologic examinations (upper gastrointestinal series) for adults over 40 years of age free of charge. The lifetime gastric cancer screening rate increased from 52% in 2004 to 85.5% in 2018 among recommended adults in South Korea [3]. With the increasing rate of gastric cancer screening, the detection of EGC has consistently increased, and EGC accounts for 61.0% of all gastric cancer cases in South Korea [4]. The overall survival and disease-specific survival rates of patients with EGC who undergo ESD for absolute and expanded indications have been reported to be comparable with those of patients undergoing surgery [1,5]. However, higher incidence rates of local recurrence, metachronous cancer, and synchronous cancer are disadvantages of ESD compared with surgical resection of EGC. Therefore, strict and careful endoscopic surveillance is recommended, even if the EGC lesion has been resected curatively. An annual endoscopic examination is usually needed to detect synchronous, metachronous, and local recurrent lesions in the remaining stomach [6].

With an increasing number of cases with endoscopically resected EGC lesions, the detection of locally recurrent lesions after endoscopic resection is a concern during follow-up surveillance. Lesions that develop on or adjacent to the previous artificial ulcer scar are difficult to resect endoscopically because of extensive submucosal fibrosis. If the recurrent lesion is premalignant, such as low- or high-grade dysplasia, additional ESD or destructive therapy using argon plasma coagulation is feasible. However, if invasive carcinoma is suspected, surgical gastrectomy should be considered first because the risk for incomplete resection is higher for recurrent lesions than for naive EGC, and remnant invasive cancer could be related to lymph node metastasis in the future. Therefore, the prediction of the risk of local recurrence after ESD is important in the management of patients with EGC.

In the present study, we evaluated the clinical and endoscopic characteristics of EGC in patients who underwent ESD to analyze the risk factors associated with local recurrence after ESD.

## 2. Materials and Methods

### 2.1. Patients

The medical records of patients who were consecutively admitted between November 2008 and February 2016 and underwent ESD at Pusan National University Yangsan Hospital in the Republic of Korea were retrospectively reviewed. During the study period, 2138 consecutive lesions were removed using the ESD method. The exclusion criteria were as follows: low-grade dysplasia (n = 1022), high-grade dysplasia (n = 188), hyperplastic polyp (n = 1), inflammatory fibrinoid polyp (n = 9), leiomyoma (n = 1), ectopic pancreas (n = 11), duplication cyst (n = 1), lipoma (n = 1), fibrous tumor (n = 1), gastritis cystica profunda (n = 5), neuroendocrine tumor (n = 4), follicular lymphoma (n = 1), carcinoma with lymphoid stroma (n = 3), immediate surgical resection after ESD (n = 18), no evidence of tumor after ESD (n = 39), and no follow-up endoscopic examination for more than 1 year (n = 192). After exclusion, 641 patients with EGC were enrolled and analyzed (Figure 1).

### 2.2. ESD Procedure and Follow-Up Examinations

All ESD procedures were performed under conscious sedation using intravenous pethidine (50 mg) and midazolam (0.05 mg/kg). ESD was performed using a needle or an insulation-tipped electrosurgical knife. All lesions were marked at 1–2 mm outside the lesion, and a mucosal incision was made outside the marking. Subsequently, submucosal dissection was performed using the same type of knives. The submucosal injection solution was a mixture of normal saline with epinephrine and indigo carmine. During or after complete resection of the lesion, preventive coagulation was performed for all exposed vessels in the artificial ulcer bed.

Follow-up endoscopic examinations were recommended at 3–6 months after the initial ESD. After the first follow-up examination, regular gastric cancer screening examination, including abdominal computed tomography and endoscopic examination, was recommended at 6–12-month intervals for at least 5 years. Endoscopic forceps biopsy of the scar after ESD was performed 1–2 times during follow-up endoscopic examinations. For patients with local recurrence, we recommended additive treatments such as surgical gastrectomy, ESD, and argon plasma coagulation.

### 2.3. Definitions

All endoscopic photographs during ESD and patient characteristics were reviewed by three endoscopists (CW Choi, SJ Kim, and DG Ryu). If the descriptions of the lesions were inconsistent, the photographs were reviewed together until agreement was achieved. The location of the lesion was classified as the lower third, mid-third, or upper third of the stomach according to the Japanese classification of gastric cancer [6]. The maximum diameter of the resected lesion was estimated after the pathological examination. The time of the procedure was estimated as the time from marking around the lesion to the end of preventive coagulation of the artificial ulcer bed after tumor resection. The present study used the Paris classification to describe the gross appearance of EGC lesions based on endoscopic findings (elevated, flat, and depressed) [7].

The surface changes of EGC lesions were determined by comparing them with the surrounding normal mucosa. Erythematous color changes were defined as the presence of reddish-colored mucosa compared with the surrounding mucosa. Surface nodularity was defined as the presence of irregularly raised or nodular mucosa. A simple mucosal defect was described as an erosion. Active deep mucosal or submucosal defects observed during the endoscopic examination were described as active ulcers. Scar deformity was defined as definite mucosal scarring or mucosal changes such as clubbing or tapering of the mucosa. Submucosal fibrosis was defined if it was visible during the ESD procedure (Figure 2). Local recurrence was defined as the presence of neoplastic lesions (adenomatous and EGC lesions) that were discovered at or adjacent to the site of the post-ESD scar (Figure 3).

The present study used the Japanese classification of gastric carcinoma to determine the histologic diagnosis as well-differentiated or undifferentiated adenocarcinoma [6]. The removed specimens were stretched, pinned, and fixed in formalin. Specimens that were resected in a piecemeal fashion were reconstructed as accurately as possible. Fixed specimens were then sectioned at 2-mm intervals. Endoscopic resection of a lesion as one piece was defined as en bloc resection. Endoscopic histologic complete resection was defined as the absence of tumor cells at the margin of the en bloc-resected specimen.

### 2.4. Statistical Analysis

Univariate analyses using a chi-square test or Fisher’s exact test for categorical variables and a Student’s t-test for continuous variables were performed. Multivariate analysis using multiple logistic regression models was performed for characteristics. A *p*-value < 0.05 was considered statistically significant. Statistical calculations were performed using the Statistical Package for the Social Sciences (SPSS) version 21.0 for Windows (IBM Corp., Armonk, NY, USA).

## 3. Results

### 3.1. Baseline Characteristics of Patients Who Underwent ESD for EGC

A total of 641 patients were enrolled in the study. The patients were predominantly men (77.2%), with a mean age of 69.3 ± 9.5 years. Most EGC lesions were located in the lower third of the stomach (70.2%). The mean tumor size was 14.1 ± 8.7 mm. The mean procedure time was 26.5 ± 17.9 min. The mean follow-up period after ESD was 50.7 ± 32.5 months. The most common gross type was the depressed type (59.8%). The en bloc resection and complete resection rates were 97.8% and 93.6%, respectively (Table 1).

### 3.2. Characteristics Associated with Local Recurrence after ESD for EGC

Local recurrence was noted in 20 patients (3.1%) (Table 1). After ESD for EGC, the tumor size, procedure time, piecemeal resection, incomplete histologic resection, undifferentiated histology, submucosal fibrosis, and surface configuration changes (erythema, nodularity, erosions, ulceration, and scar) were different between the recurrence and no evidence of recurrence groups (Table 1).

On multivariate analysis, lesion size ≥ 15 mm (odds ratio [OR], 8.186; 95% confidence interval [CI], 1.725–38.838, *p* = 0.008), incomplete pathologic resection (OR, 11.518, 95% CI, 2.997–44.268; *p <* 0.001), undifferentiated carcinoma (OR, 5.580; 95% CI, 1.120–27.785; *p* = 0.036), scar deformity (OR, 7.222; 95% CI, 1.350–38.616; *p* = 0.021), and no evidence of surface erythema (OR, 23.014; 95% CI, 4.674–113.315; *p* < 0.001) were associated with local recurrence (Table 2).

### 3.3. Clinical Outcomes after Local Recurrence

Among cases of complete histologic resections (n = 610), 11 had local recurrence (1.8%). Two out of seven patients who underwent additional surgical gastrectomy showed lymph node metastasis. Three patients were treated with additive ESD. One patient who had deep submucosal invasive cancer with lymphatic invasion and refused additive surgical resection after ESD showed local recurrence and hepatic metastasis 2 years after ESD (Figure 4a).

Among cases of incomplete histologic resections (n = 49), 11 had local recurrence (22.4%). Among the five patients who underwent additive operations, one revealed lymph node metastasis. Three patients who underwent additional ESD and one patient who received endoscopic destructive therapy with argon plasma coagulation showed no evidence of recurrence during the follow-up period (60–136 months). Two patients refused additional surgical or endoscopic therapy, and we were unable to follow up with these patients (Figure 4b).

## 4. Discussion

The present study shows that larger lesion size, incomplete pathologic resection, undifferentiated carcinoma, scar deformity, and no evidence of surface erythema were associated with local recurrence after ESD for EGCs. Endoscopic resection of EGC without lymph node metastasis has been accepted as a definite treatment method with long-term outcomes comparable to those of surgical resection of EGC. However, some lesions show local recurrence at the artificial ulcer scar after ESD. One drawback of endoscopic resection compared with surgical resection is the higher rate of local recurrence [1]. There has been no definite recommendation on whether to select surgical resection or endoscopic treatment for local recurrence of EGC after endoscopic resection. To date, additive surgical resection of locally recurrent EGC lesions is recommended first because the additional endoscopic curative resection of locally recurred EGC lesions is more difficult than that of naive ESD lesions because of widespread submucosal fibrosis. However, some patients in the older age group with poor performance status who refuse to undergo surgical resection could be candidates for endoscopic treatment.

In the present study, the proportion of locally recurrent cases was 3.1%. The reported rates of locally recurrent EGC after ESD range from 0.7% to 3.7% [8,9]. We defined a locally recurred lesion as the neoplastic lesion that developed at the post-ESD artificial ulcer scar. Some recurrent lesions may be metachronous or synchronous lesions in relation to the mucosa near the artificial ulcer scar. On endoscopic photo review, nine lesions were found mainly at the margin of the artificial ulcer. If the recurred lesions are located mainly at the margin of the tumor, endoscopic resection may be preferred over surgical resection. We performed successful additive endoscopic resection in six patients. If the recurred lesions are located mainly at the center of the scar, surgical resection is preferred. Twelve patients received surgical resections with lymph node dissection (three patients showed lymph node metastasis), and one patient who refused surgical resection underwent argon plasma destructive therapy. During the period of the study, only one case of gastric cancer-related death was found where the patient had refused additional surgical resection for deep submucosal and lymphovascular invasive EGC after ESD. Therefore, if we had kept endoscopic treatment indications for EGC, the number of gastric cancer-related deaths would have been zero during the study period.

In the present study, incomplete pathologic resection was significantly associated with local recurrence after ESD. The reported incidence of local recurrence after incomplete resection was 4.2–30% [10,11]. Incomplete resection is a result of piecemeal resection or marginal involvement of EGC after a one-piece resection. Piecemeal resection was not a significant association in the present study because most of the lesions were resected in one piece (97.8%). Only two lesions had recurred after piecemeal resection, and the macroscopic photographs showed no residual tumors in most of the piecemeal-resected specimens. In piecemeal-resected tumors, an estimation of marginal status is difficult. Therefore, after pathological confirmation of the undetermined marginal status, a clean artificial ulcer bed and resection of the mucosa outside the marking made before mucosal incision may be important. In addition to the macroscopic findings after endoscopic resection, pathologic differentiation and the depth of invasion of the resected specimen may be key factors in deciding whether to perform additional surgical resection. In our institution, additional operative treatment is recommended for piecemeal-resected tumors under the following conditions: remnant gastric cancer is highly suspected macroscopically after endoscopic resection (irregular ulcer bed or mucosal incision performed inside the mucosal marking during ESD) or undifferentiated histology, submucosal invasion, or lymphatic invasion is observed in the resected specimen. If the piecemeal-resected EGC lesion shows well-differentiated carcinoma and mucosal cancer in the resected specimen and an artificial ulcer bed and margin with no visible evidence of remnant EGC, a discussion with the multidisciplinary team, including gastroenterologists and the surgical team, is undertaken to decide whether to perform surgical resection. Marginal status is important in predicting local recurrence. Since surgical resection was recommended for all the vertical marginal-positive patients in the present study, the lateral marginal status might be an important predictive factor. A previous study reported that a longer length of the involved lateral margin was important for predicting local recurrence [12].

Undifferentiated adenocarcinoma was associated with local recurrence after ESD. Undifferentiated gastric tumor cells originating in the neck of the gastric gland can spread to the submucosal space [13]. Therefore, an estimation of the resection margin of undifferentiated EGC during ESD is more difficult than that of differentiated EGC. Moreover, remnant EGC is not predictable based on endoscopic findings. In the present study, surface erythema was not a risk factor for local recurrence. Surface erythema is an important endoscopic finding associated with well-differentiated EGC [14]. In contrast, undifferentiated EGC such as signet-ring cell carcinoma invades and spreads underneath the surface epithelium without destruction of the mucosal epithelium and subepithelial capillaries; therefore, no surface erythema is observed during endoscopic examination in these cases [15]. In addition, EGC with fibrotic submucosa or a scar does not show erythema. EGC without surface erythema is more likely to be diagnosed as undifferentiated gastric cancer or submucosal invasive cancer that is considered to have a higher risk of recurrence. Moreover, in the absence of erythema, it is difficult to distinguish the exact margins, and this can increase the risk of incomplete resection and recurrence.

Larger lesion size is an important factor for local recurrence after ESD. Because endoscopic snare has a size limitation, tumors more than 20 mm in size are difficult to resect by simple snaring with a safe lateral margin. Although ESD is a more effective treatment method regardless of tumor size than conventional EMR, a large tumor is more difficult to resect safely than a small tumor [11]. Therefore, larger tumor size is highly associated with incomplete endoscopic resection of tumors. In addition, ulcer scar formation is associated with local recurrence. Because ulcer deformity represents extensive submucosal fibrosis, exact submucosal dissection beneath the EGC is difficult without direct visualization of the submucosal layer. In addition, if the EGC has existing deep ulceration, the estimation of the vertical marginal status might be inaccurate. The possible presence of remnant cancer cells beneath the dissected ulcer scar could be a risk factor for local recurrence.

Incomplete resection of the tumor is highly associated with procedural difficulty. Difficult ESD procedures are associated with several factors, such as larger lesion size, location of the tumor, which is difficult to reach using endoscopic electrosurgical knives, submucosal fibrosis, submucosal invasive cancer, and the expertise of endoscopists [16]. It is challenging to predict the difficulty of the procedure because all procedural situations are not the same. When a difficult endoscopic procedure is expected, preparations for all possible methods to overcome the difficult situation should be considered, including consultations with more experienced endoscopists.

The present study had several limitations. First, because the present study was retrospective in a single referral medical cancer, there was an inherent selective bias. Second, the sample of locally recurrent tumors might be too small to analyze the factors associated with local recurrence and generalize the results. However, the results of this study are informative and consistent with those of other studies [8,9]. Third, some of the locally recurrent lesions might be metachronous or missed synchronous lesions, which could not be differentiated using retrospective endoscopic photograph reviews. Fourth, the length of submucosal fibrosis or the shape of the scar might be an important factor associated with the difficulty of ESD. A short length of submucosal fibrosis might be associated with repeated endoscopic forceps biopsies. However, because of the retrospective study design, we were unable to estimate the length of submucosal fibrosis. Therefore, further prospective studies should be considered. However, additive information of this present study would be helpful to the clinical practice after ESD for EGCs.

## 5. Conclusions

In conclusion, even though local recurrence after ESD of EGC is rarely observed, the possibility of its recurrence is not negligible. Furthermore, local recurrence can present after complete endoscopic resection of EGC lesions. We should check for synchronous, metachronous, and locally recurrent lesions that might be associated with larger lesion size, incomplete histological resection, undifferentiated cancer, scar formation, and the absence of surface erythema. The results of this study will be helpful for the effective detection and management of local recurrences of EGC after ESD.

## Figures and Tables

**Figure 1 jcm-12-02018-f001:**
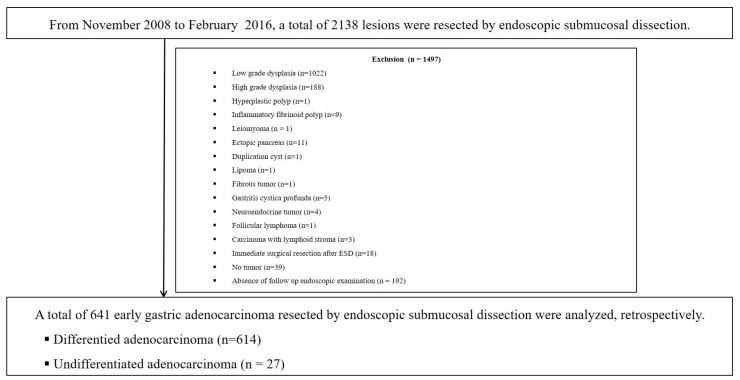
Schematic showing a flow chart of the study. ESD, endoscopic submucosal dissection.

**Figure 2 jcm-12-02018-f002:**
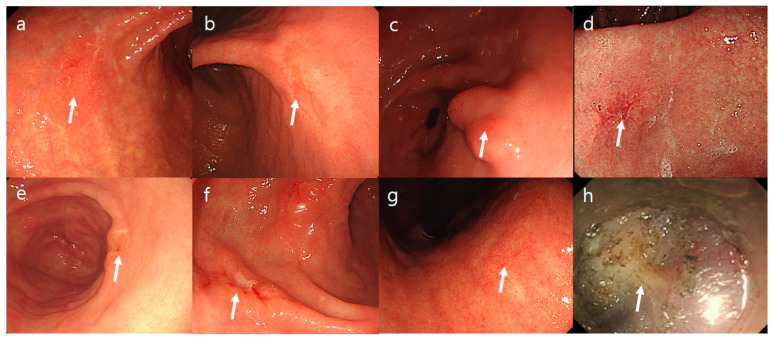
Endoscopic characteristics of early gastric cancer lesions. (**a**,**b**) Erythematous mucosa. (**c**) Nodular mucosa. (**d**) Depression. (**e**) Erosion. (**f**) Active ulcer. (**g**) Scar. (**h**) Submucosal fibrosis.

**Figure 3 jcm-12-02018-f003:**
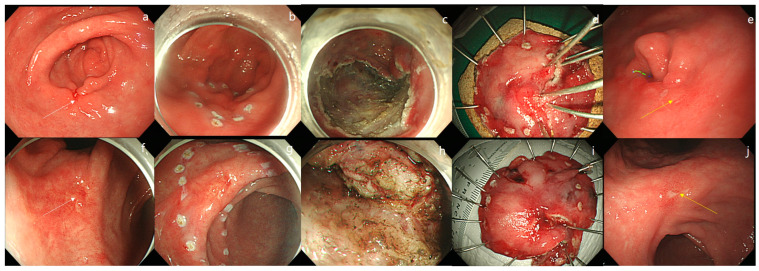
Endoscopic findings of early gastric cancer (EGC) with local recurrence. (**a**) Local recurrence at the proximal margin of the artificial ulcer scar. (**b**,**c**) A 20-mm sized, well-differentiated EGC lesion with ulceration was resected using the endoscopic submucosal dissection (ESD) method. (**d**) Although the artificial ulcer was clear, the lesion was resected in two pieces. Pathologic examination revealed mucosal cancer without lymphatic invasion and negative lateral margin. (**e**) After 8 years, an 18-mm sized, well-differentiated EGC lesion is observed. (**f**) Local recurrence at the center of the artificial ulcer scar. (**g**) A 22-mm sized, well-differentiated EGC lesion is observed with a suspicious ulcer scar deformity. (**h**) Extensive submucosal fibrosis is observed. (**i**) ESD was performed en-bloc. (**j**) After two years, a 20-mm sized, recurrent EGC lesion with ulceration is observed.

**Figure 4 jcm-12-02018-f004:**
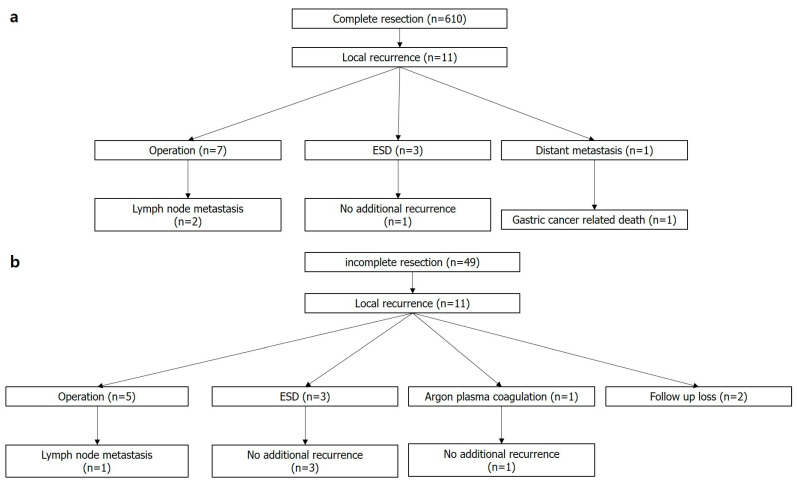
Clinical outcomes of patients with locally recurrent lesions. (**a**) Complete resection, (**b**) Incomplete resection. ESD, endoscopic submucosal dissection; n, number.

**Table 1 jcm-12-02018-t001:** Baseline characteristics of early gastric cancer patients at the time of initial ESD and comparison between the presence of local recurrence and the absence of local recurrence.

	No Evidence of Local Recurrence (n = 621)	Evidence of Local Recurrence (n = 20)	Total (n = 641)	*p* Value
Age, years, mean ± SD	69.4 (9.5)	66.2 (9.6)	69.3 (9.5)	0.144
Age ≥ 65 years	438 (70.5)	12 (60.0)	450 (70.2)	0.311
Male, Sex, n (%)	478 (77.0)	17 (85.0)	495 (77.2)	0.399
Tumor location, n (%)				0.103
Lower third	440 (70.9)	10 (50.0)	450 (70.2)	
Middle third	130 (20.9)	8 (40.0)	138 (21.5)	
Upper third	51 (8.2)	2 (10.0)	53 (8.3)	
Lesion size (mm, mean ± SD)	13.9 (8.6)	22.2 (8.2)	14.2 (8.7)	<0.001
Lesion size (Maximal diameter) > 15 mm, n (%)	227 (36.6)	17 (85.0)	244 (38.1)	<0.001
Procedure time (min, mean ± SD)	26.2 (17.7)	36.1 (21.1)	26.5 (17.9)	0.015
Procedure time ≥ 30 min	214 (34.5)	12 (60.0)	226 (35.3)	0.019
Follow-up in months, mean (SD)	50.3 (32.5)	64.6 (31.1)	50.7 (32.5)	0.144
Lymph node metastasis, n (%)	0 (0)	4 (20.0)	4 (0.6)	<0.001
En bloc resection, (n, %)	610 (98.2)	17 (85.0)	627 (97.8)	<0.001
Complete resection, (n, %)	590 (95.0)	10 (50.0)	600 (93.6)	<0.001
Pathologic diagnosis (n, %)				0.015
Differentiated adenocarcinoma	597 (96.1)	17 (85.0)	614 (95.8)	
Undifferentiated adenocarcinoma	24 (3.9)	3 (15.0)	27 (4.2)	
Gross type, n (%)				0.277
Elevated	191 (30.8)	5 (25.0)	196 (30.6)	
Flat	58 (9.3)	4 (20.0)	62 (9.7)	
Depressed	372 (59.9)	11 (55.0)	383 (59.8)	
Surface configuration, n (%)				
Erythema	595 (95.8)	15 (75.0)	610 (95.2)	<0.001
Nodularity	170 (27.4)	10 (50.0)	180 (28.1)	0.027
Depression	283 (45.6)	11 (55.0)	294 (45.9)	0.405
Erosion	279 (44.9)	4 (20.0)	283 (44.1)	0.027
Active Ulcer	46 (7.4)	4 (20.0)	50 (7.8)	0.039
Scar	97 (15.6)	12 (60.0)	109 (17.0)	<0.001
Submucosal fibrosis, n (%)	193 (31.1)	14 (70.0)	207 (32.3)	<0.001
Invasion depth, (n, %)				0.080
Mucosa	545 (87.8)	16 (80.0)	561 (87.5)	
SM1	36 (5.8)	2 (10.0)	38 (5.9)	
SM2	13 (2.1)	2 (10.0)	15 (2.3)	
SM3	27 (4.3)	0 (0)	27 (4.2)	
Lymphovascular invasion, n (%)	15 (2.4)	1 (5.0)	16 (2.5)	0.466
Delayed bleeding, n (%)	44 (7.1)	2 (10.0)	46 (7.2)	0.619
Latrogenic perforation, n (%)	4 (0.6)	0 (0)	4 (0.6)	0.719

ESD, endoscopic submucosal dissection; n, number; SD, standard deviation; SM1, submucosal invasion <500 µm; SM2, submucosal invasion 500–1000 µm; SM3, submucosal invasion >1000 µm.

**Table 2 jcm-12-02018-t002:** Multivariate analysis of factors associated with local recurrence.

Results at the Time of Initial ESD	Adjusted OR (95% CI)	*p* Value
Lesion size > 15 mm	8.186 (1.725–38.838)	0.008
Procedure time > 30 min	1.477 (0.427–5.105)	0.538
Incomplete histologic resection	11.518 (2.997–44.268)	<0.001
Piecemeal resection	1.212 (0.176–8.346)	0.845
Undifferentiated carcinoma	5.580 (1.120–27.785)	0.036
Scar	7.222 (1.350–38.616)	0.021
Submucosal fibrosis	1.255 (0.210–7.502)	0.803
Without erythema	23.014(4.674–113.315)	<0.001
Erosion	1.872 (0.473–7.400)	0.372
Active ulcer	1.102 (0.238–5.092)	0.900

ESD, endoscopic submucosal dissection; CI, confidence interval; OR, odds ratio.

## Data Availability

We uploaded the data of our study on the website: https://github.com/gon22gon/EGC_local-recur (accessed on 23 December 2021) for the purpose of academic sharing.

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
