# Peer review of "Local Recurrence after Endoscopic Submucosal Dissection of Early Gastric Cancer"

_jcm, 2023, doi:10.3390/jcm12052018_

Round 1

Reviewer 1 Report

Your article "Local recurrence after endoscopic submucosal dissection of 2 early gastric cancer" is interesting. A more detailed description of the methods and results is recommended. In addition, it is necessary to cite a more up-to-date bibliography. I recommend the english editing, especially of the conclusion.

Reviewer 2 Report

The authors have conducted a thorough analysis of a large dataset spanning eight years, and have presented important findings pertaining to local recurrence and its associated factors. However, there are a few points that may benefit from further clarification or elaboration:

1. In Figure 4, for the group of patients who underwent incomplete resection, the number of patients who received ESD treatment is stated as n=3, but the number of patients who did not experience additional recurrence is stated as n=4. It may be helpful to clarify whether this discrepancy is due to an error or a specific reason.

2. The manuscript indicates that the rate of lymph node metastasis among patients who underwent surgery after local recurrence is relatively high, at 28.6% in the complete resection group and 20% in the incomplete resection group. It would be useful to know if there was a specific protocol in place for determining whether patients should receive repeat ESD or surgery, and if this protocol took into account the risk of lymph node metastasis.

3.  The manuscript notes that no additional recurrence was observed among patients who received repeat ESD and Argon plasma treatment, but it would be helpful to know the actual duration of follow-up in these cases.

4. The authors state that no evidence of surface erythema was associated with local recurrence, however, Table 1 suggests that 75% of patients with local recurrence did in fact present with erythema. While this rate is lower than the 95.8% of patients without recurrence, it is still worth noting that many cases of local recurrence do present with erythematous changes.

5. The authors propose that ESD scar without erythematous change may be associated with local recurrence. This is a unique and interesting finding, but it would be beneficial to have more information to support this conclusion. For example, it would be helpful to see visual examples of scars without erythematous change, and to have information about inter-observer agreement for the identification of erythematous change to boost the credibility of the conclusion
